# In-depth single-cell analysis of translation-competent HIV-1 reservoirs identifies cellular sources of plasma viremia

Basiel Cole[1], Laurens Lambrechts [1,2], Pierre Gantner[3], Ytse Noppe[1], Noah Bonine[1,2], Wojciech Witkowski[1], Lennie Chen[4], Sarah Palmer[5], James I. Mullins[4,6,7], Nicolas Chomont [3,8], Marion Pardons[1,9] & Linos Vandekerckhove [1,9]✉

Clonal expansion of HIV-infected cells contributes to the long-term persistence of the HIV reservoir in ART-suppressed individuals. However, the contribution from cell clones that harbor inducible proviruses to plasma viremia is poorly understood. Here, we describe a single-cell approach to simultaneously sequence the TCR, integration sites and proviral genomes from translation-competent reservoir cells, called STIP-Seq. By applying this approach to blood samples from eight participants, we show that the translation-competent reservoir mainly consists of proviruses with short deletions at the 5'-end of the genome, often involving the major splice donor site. TCR and integration site sequencing reveal that cell clones with predicted pathogen-specificity can harbor inducible proviruses integrated into cancer-related genes. Furthermore, we find several matches between proviruses retrieved with STIP-Seq and plasma viruses obtained during ART and upon treatment interruption, suggesting that STIP-Seq can capture clones that are responsible for low-level viremia or viral rebound.

[1] HIV Cure Research Center, Department of Internal Medicine and Pediatrics, Ghent University Hospital, Ghent University, Ghent, Belgium. [2] BioBix, Department of Data Analysis and Mathematical Modelling, Faculty of Bioscience Engineering, Ghent University, Ghent, Belgium. [3] Department of Microbiology, Infectiology and Immunology, Université de Montréal, Montreal, QC, Canada. [4] Department of Microbiology, University of Washington, Seattle, WA, USA. [5] Centre for Virus Research, The Westmead Institute for Medical Research, The University of Sydney, Sydney, NSW, Australia. [6] Department of Medicine, University of Washington, Seattle, WA, USA. [7] Department of Global Health, University of Washington, Seattle, WA, USA. [8] Centre de Recherche du Centre Hospitalier de l'Université de Montréal, Montreal, QC, Canada. [9] These authors contributed equally: Marion Pardons, Linos Vandekerckhove. ✉email: linos.vandekerckhove@ugent.be

HIV-1 infection remains incurable due to the establishment of a persistent viral reservoir, which is unaffected by anti-retroviral therapy (ART)[1–3]. This reservoir mainly consists of long-lived memory CD4 T cells harboring latent, replication-competent proviruses, capable of refueling viremia upon treatment interruption (TI)[4,5]. The viral reservoir is remarkably stable, with an estimated half-life of ~44–48 months, suggesting that at least 70 years of continuous ART would be required to eliminate it completely[6,7]. Long-term maintenance of the reservoir can in part be explained by clonal expansion of HIV-infected cells, which is thought to be driven by three non-mutually exclusive forces: homeostatic proliferation[8–12], antigenic stimulation[13–15], and integration site (IS)-driven proliferation[16–19]. Identifying the cellular sources of viral rebound and the mechanisms that ensure their persistence during ART is needed to develop targeted strategies to eradicate or control HIV[20,21].

Several sequencing-based assays have been developed to study the HIV reservoir, each focusing on different aspects of the infected cells and the proviruses within. Near full-length (NFL) provirus sequencing enables the identification of genome-intact and potentially replication-competent proviruses[22–25]. Integration site analysis (ISA) pinpoints the chromosomal location of proviruses and is frequently used as a marker to study clonal expansion of infected cells[16,17]. More recently, NFL provirus sequencing and ISA were combined into a single assay, allowing the study of the relationship between proviral IS and genome structure[26–28]. However, because these assays are usually performed on bulk CD4 T cell DNA, they mainly identify defective proviruses, as it has been estimated that only 2–5% of the total proviruses are genome-intact[22,29,30]. As such, they do not focus on proviruses that could lead to viral rebound upon TI. On the contrary, viral outgrowth assays (VOA) combined with NFL viral genome sequencing enable the characterization of replication-competent proviruses[3,6,31]. However, the IS of the provirus as well as the phenotype and TCR sequence of the infected cell cannot be determined with this assay.

Alternative assays have been developed to characterize and quantify infected cells harboring transcription-competent[19,32] or translation-competent[14,32–36] proviruses, therefore enriching for proviruses with a higher probability of contributing to viral rebound[37]. These assays use a potent stimulant to reactivate proviruses from latency, inducing transcription of viral genes and production of viral proteins. Infected cells can then be identified and isolated by fluorescence-activated cell sorting (FACS). This allowed for the characterization of NFL proviral genome structure[19,35], TCR sequences[14,35], and IS[19] from cells harboring an inducible provirus. However, none of these methodologies capture all three layers of information simultaneously.

Here, we present a novel method, called HIV STIP-Seq: Simultaneous TCR, Integration site and Provirus sequencing. STIP-Seq enables sequencing of the proviral genome and matched IS of translation-competent proviruses, as well as phenotypic characterization and TCR sequencing of the host cell. We used this approach to characterize infected cells that harbor inducible proviruses from eight individuals on suppressive ART. Furthermore, three out of eight participants underwent an ATI, which allowed us to investigate the contribution of the translation-competent reservoir to residual viremia and viral rebound.

## Results

**STIP-Seq.** STIP-Seq is a derivative of the HIV-Flow assay[33], with the addition of downstream whole genome amplification (WGA) and sequencing of the provirus, IS and TCR. Since WGA by multiple displacement amplification (MDA) is not compatible with cross-linking fixatives such as paraformaldehyde, we used methanol for simultaneous fixation and permeabilization, permitting efficient amplification of the cellular genome. Using a dilution series of J1.1 cells in the parental Jurkat cell line, we showed good linearity of the frequency of p24+ cells assessed by the methanol-based HIV-Flow assay, down to ~3 p24+ cells/million cells ($R^2 = 0.99$ and Supplementary Fig. 1a, b). In addition, methanol fixation did not have a significant impact on the frequency of p24+ cells ($p = 0.84$, Supplementary Fig. 1c).

Following methanol-based HIV-Flow, p24+ cells were sorted into individual wells of a 96-well plate (Fig. 1a, b). Single-cell WGA by MDA was used to amplify the DNA of single-sorted p24+ cells, including the provirus integrated within. Amplified genomes were subjected to ISA by Integration Site Loop Amplification (ISLA) and NFL proviral sequencing using either a 5-, 4-, or 2-amplicon PCR approach (Fig. 1a and Supplementary Fig. 2). In addition, the TCRβ chain of the host cell was sequenced as described[14], and the memory phenotype of the cell was determined post hoc by index sorting, during which the level of expression of all phenotypic markers on single-sorted cells was recorded (Fig. 1a and Supplementary Fig. 2).

**Provirus characterization of p24-producing cells with STIP-Seq.** To investigate the characteristics of p24-producing cells and their associated proviruses, we performed STIP-Seq on single-sorted CD4 T cells from eight ART-suppressed individuals (Supplementary Table 1). Of note, for participant P5, STIP-Seq was performed on two samples collected 3 years apart. A total of 158 p24+ cells and 156 IS were retrieved. A large proportion of these stemmed from clonally expanded infected cells (74%, 116/156), defined by recurrent identical IS, confirming the often clonal nature of the translation-competent reservoir[14].

NFL proviral genome sequencing yielded a total of 40 distinct genomes with complete coverage, which fell within one of three categories: genome-intact (12.5%, 5/40), packaging signal (PSI) and/or major splice donor (MSD) defects (85%, 34/40), or large internal deletion (2.5%, 1/40) (Fig. 2a and Supplementary Table 2). The PSI/MSD defective proviruses usually had deletions spanning one or more packaging stem-loops, all of them involving the MSD located within stem–loop 2 (Fig. 2b). Among these, we identified seven proviruses with deletions covering the binding region of the forward primer from the 5-amplicon NFL PCR (U5-638F), although these deletions could be spanned by the two-amplicon approach (F581; Fig. 2b, indicated with triangles). Intriguingly, 16 proviruses (40%) had PSI/MSD deletions extending into the p17 gene, removing the AUG start codon of the Gag polyprotein (Supplementary Fig. 3). This implies the use of an alternative start codon to enable the translation of the p24 protein[38]. Of note, p24 antibody fluorescence intensities did not differ between cells harboring Gag AUG-defective vs. Gag AUG-intact proviruses, indicating that both groups of cells produce comparable amounts of p24 protein upon PMA/ionomycin stimulation ($p = 0.71$ for APC, $p = 0.70$ for FITC, Supplementary Fig. 4a–c). Similarly, no significant difference in p24 fluorescence intensity was observed between cells harboring NFL genome-intact vs. defective proviruses ($p = 0.70$ for APC, $p = 0.49$ for FITC, Supplementary Fig. 4d, e). Out of 40 distinct NFL sequences analyzed, only 1 had a large internal deletion (1191 bp; Fig. 2a), and none displayed inversions or hypermutations (Fig. 2a). This is in contrast with previously reported NFL data that were generated on bulk CD4 T cell DNA[22–24,29]. To investigate this disparity, we compared NFL genomes obtained by Full-Length Individual Provirus Sequencing (FLIPS)[22] on bulk CD4 T cell DNA with NFL sequences retrieved by STIP-Seq, for two longitudinal samples from participant P5 (Supplementary Fig. 5). This analysis showed that proviruses with large internal

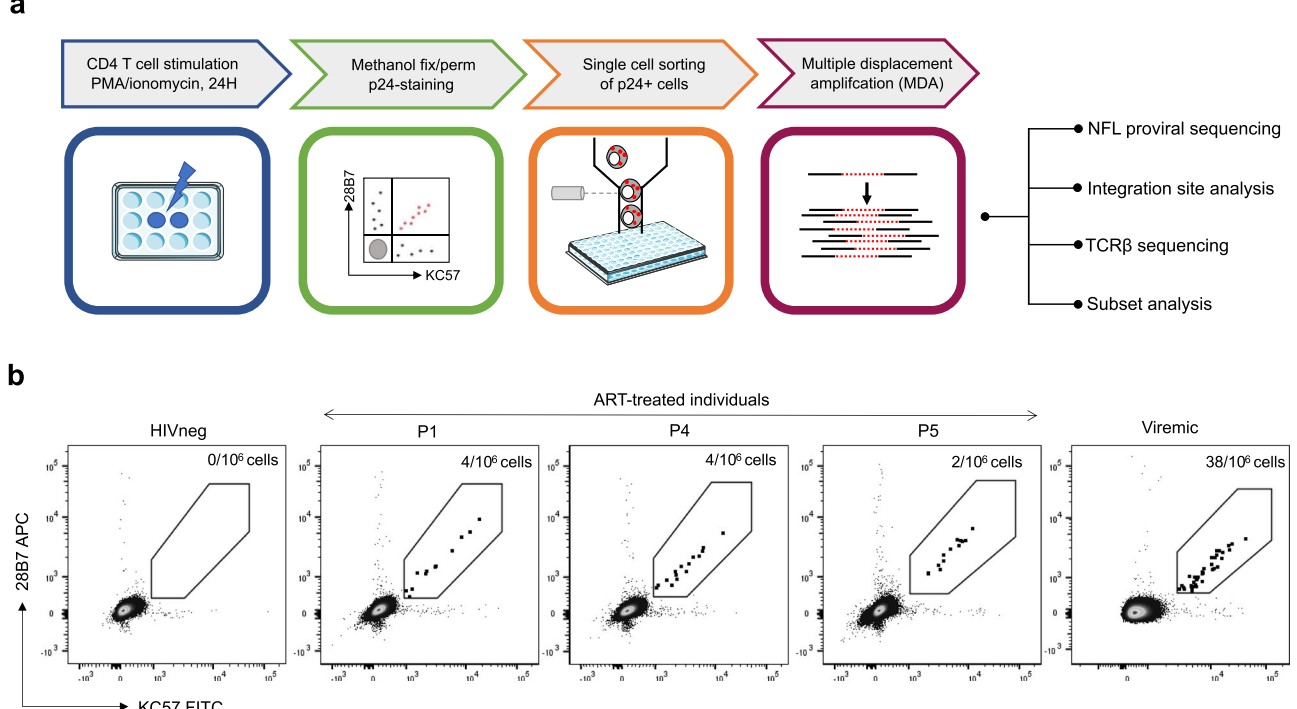

**Fig. 1 STIP-Seq enables isolation and characterization of p24-producing cells after PMA/ionomycin stimulation. a** Overview of the STIP-Seq assay. CD4 T cells are stimulated for 24 h with PMA (162 nM)/ionomycin (1 μg/mL). Cells are fixed and permeabilized with methanol, and p24-producing cells are identified using a combination of two antibodies (KC57 and 28B7) targeting the p24 protein. p24+ cells are single-cell sorted by flow cytometry. DNA from p24+ cells is amplified by multiple displacement amplification, before performing near full-length (NFL) proviral genome sequencing, integration site analysis, TCR sequencing, and post hoc determination of the CD4 T cell memory phenotype. The illustrations of the 96-well plates were obtained from SMART (Servier Medical Art; http://smart.servier.com/), licensed under a Creative Common Attribution 3.0 Unported license (https://creativecommons.org/licenses/by/3.0/). NFL near full-length. **b** Representative FACS dot plots showing the KC57-FITC/28B7-APC co-staining on CD4 T cells from one HIV non-infected control, one viremic, and three ART-treated individuals.

deletions and hypermutations were absent in p24+ cells, although highly prevalent in bulk CD4 T cells (1/65 hypermutated, 58/65 with a large internal deletion, 4/65 with a PSI/MSD defect, Supplementary Fig. 5). We acknowledge the possibility that the number of truncated proviruses recovered by STIP-Seq might be underestimated since we used multi-amplicon approaches to amplify the proviral genomes, which could preclude the detection of truncated proviruses that contain deletions in the primer binding sites regions. At the second time point, 2/31 proviral genomes recovered from bulk CD4 T cells were intact, whereas none were detected in p24+ cells (0/11) (Supplementary Fig. 5). This suggests that these proviruses were not induced by a single round of PMA/ionomycin stimulation, or they were missed due to the more limited sampling with STIP-Seq. Finally, we did not observe a significant difference in NFL class proportions between unique proviruses and those stemming from clonal expansions, suggesting that cellular proliferation does not select for a specific type of genomic structure ($p = 0.99$, Supplementary Fig. 6).

In order to link the chromosomal location of proviruses to their corresponding genome structure, ISA was performed on successfully amplified genomes. A bias towards integration in the reverse orientation with respect to the gene was observed (36/58 in reverse orientation, 12/58 in same orientation, 3/58 in region with gene on either strand, 9/58 in intergenic region) (Supplementary Table 3). Previous studies have shown an enrichment of IS in cancer-associated genes, such as *STAT5B* and *BACH2*, suggesting IS-driven expansion of infected cells[16–18]. Out of 58 distinct IS, 11 were located within genes associated with cellular proliferation (Fig. 2a and Supplementary Table 3, indicated with asterisks). When restricting the analysis to distinct clonal cell populations, 27%

(6/22) had an integrated provirus in a gene associated with cellular proliferation. This frequency is similar to those observed by Wagner et al.[16] (30%, 9/30) and Maldarelli et al.[17] (58%, 17/29) on bulk CD4 T cell DNA, suggesting that STIP-Seq does not enrich for cells with an integrated provirus in a gene involved in cellular proliferation. Among all participants, three different IS in the *STAT5B* gene were identified, two of which could be attributed to clonal cell populations. Of note, all three proviruses were integrated in the opposite orientation with respect to the gene. Interestingly, for participant P4, a cell with an intact provirus integrated in the *ZNF274* gene was retrieved (Fig. 2a). This gene was previously described as located in a dense heterochromatin region and was associated with proviruses in a state of "deep latency"[39].

It was previously shown that p24+ cells mainly display central memory (TCM), transitional memory (TTM) and effector memory (TEM) phenotypes[14,33]. Consistent with this, all but one of the cells identified with STIP-Seq fell within these memory subsets (60/143 TCM/TTM, 82/143 TEM), with a single-cell displaying a naïve phenotype (1/143 TN). When restricting the analysis to clones, 9/20 were found in both the TCM/TTM and the TEM subset (Fig. 2c), an observation that was previously reported[14,15]. Of note, proportions of CD4 T cell subsets were only minimally affected by a 24 h PMA/ionomycin stimulation and methanol fixation (Supplementary Fig. 7).

In conclusion, these results show that p24+ cells preferentially display a memory phenotype and are enriched in NFL proviral genomes that have deletions at the 5′ end of the genome. Our data suggest that the MSD, located within stem–loop 2, is a particular hotspot for deletion among translation-competent proviruses.

**TCRβ sequencing reveals clones with predicted pathogen-specificity**. Under the hypothesis that infected cell clones with responsiveness towards a pathogen could have arisen due to cognate antigen exposure, we attempted to predict the specificity of p24+ cells based on the CDR3 region of the TCRβ sequence, as described[14]. A total number of 43 distinct TCRβ sequences were

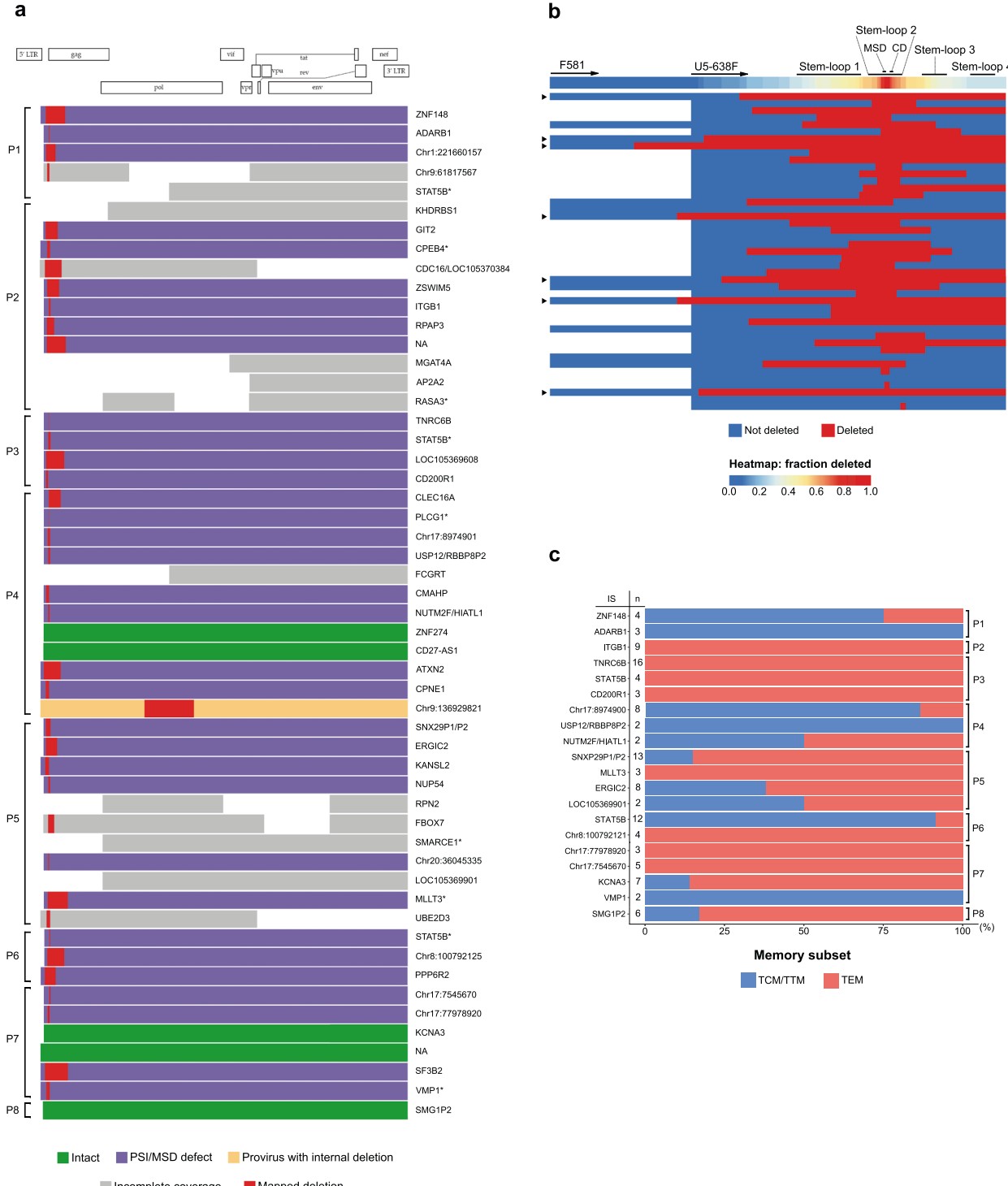

**Fig. 2 Near full-length proviral sequencing, integration site analysis, and subset analysis on p24-producing cells from ART-treated individuals.**
**a** Virogram showing the near full-length proviral genomes recovered from eight ART-treated individuals. Proviral genomes were reconstructed using a 5-amplicon, 2-amplicon, or 4-amplicon PCR approach. Corresponding integration sites (IS) are indicated at the right-hand side of each proviral genome. Cancer-related genes are indicated with an asterisk. **b** Heatmap of the deletions in the 5′ UTR region, including the Ψ packaging signal. The second-round forward primers for the 2-amplicon (F591) and 5-amplicon (U5-638F) NFL PCR approach are annotated with arrows on the heatmap. Proviruses with a deletion spanning the U5-638F primer are indicated with a triangle at the left-hand side of each provirus. MSD major splice donor, CD cryptic donor.
**c** Memory subset distribution of clonal p24-producing cells. The number of cells within each clone is indicated at the left-hand side of each horizontal bar. TCM central memory T cell, TTM transitional memory T cell, TEM effector memory T cell.

retrieved. Importantly, p24+ cells that were previously determined clonal by IS sequencing were also identified as such based on TCRβ sequences. The proportion of HIV-infected cells for which specificity could be predicted was 8/43 (19%) when considering all distinct CDR3 sequences, or 5/19 (26%) when restricting to clonal populations (Fig. 3 and Supplementary Table 4). Among all participants, predicted TCR specificities of p24+ cells were confined to CMV, *M. tuberculosis*, and influenza, suggesting that infection with or immunization against these pathogens plays a role in the maintenance of the translation-competent reservoir.

Participant P3 had a clone with a predicted cross-reactive TCR (CMV, influenza, *M. tuberculosis*), for which the provirus was integrated in *CD200R1*, a gene not known to be involved in cell proliferation (Fig. 3). Participants P4 and P6 displayed clones with predicted specificity towards *M. tuberculosis* and CMV respectively (Fig. 3). Both clones harbored a provirus integrated at an intergenic region (chr17:8974901 and chr8:100792125, respectively), suggesting that their expansion was not driven by promoter insertion (Fig. 3). In contrast, we found several cells with predicted TCR-specificity towards a pathogen which contained a provirus integrated into a gene involved in cellular proliferation, as previously described by Simonetti et al.[15]. Participant P3 harbored a clone with an IS in *STAT5B*, potentially allowing for IS-driven proliferation. Moreover, the predicted TCR specificity towards influenza suggests that the seasonal flu or vaccination might have contributed to the expansion of this clone (Fig. 3). Similarly, one expanded clone from participant P7 had predicted specificity towards *M. tuberculosis* and had an intact provirus integrated in *KCNA3*, a gene involved in T cell activation and proliferation[40]. Of note, this provirus was integrated in the same orientation as the gene, which could lead to aberrant transcription and subsequent dysregulation of *KCNA3* expression.

Finally, to investigate the dynamics of the translation-competent reservoir, we performed STIP-Seq on two longitudinal samples from P5, collected 3 years apart (Supplementary Fig. 8 and Supplementary Table 1). While the largest clone at the first time point (IS in *SNX29P1/P2*) was not retrieved 3 years later, one new clone emerged (IS in *LOC105369901*) and two clones persisted (IS in *ERGIC2* and *MLLT3*). These observations confirm that HIV-infected cell clones can persist, contract or expand over time[14,16,41].

Taken together, we show that cells with a predicted TCR specificity towards pathogens can harbor inducible proviruses that are integrated in genes associated with cellular proliferation, suggesting that antigen exposure and IS-driven mechanisms can synergize to favor the persistence of the translation-competent reservoir.

**STIP-Seq sequences match plasma viruses obtained before and during ATI.** We then investigated whether proviruses retrieved with STIP-Seq overlap with plasma virus sequences before and during an ATI. In order to trace the cellular origin of plasma viruses, we performed STIP-Seq on CD4 T cells from three participants (P6, P7, P8), both at T1 (during ART; last time point before ATI) and at T2 (During ATI; last available time point with undetectable viremia) (Fig. 4a, b). Of note, because the p24+ fraction might become dominated by actively producing proviruses rather than reactivated proviruses in the context of detectable viremia (T3/T4), STIP-seq was only performed at T1/ T2. Plasma viral sequences (V1–V3 *env*, 894 bp) from before (T1) and during the ATI (T2, T3, and T4) were aligned to trimmed NFL sequences obtained with STIP-Seq and maximum-likelihood phylogenetic trees were constructed (Fig. 5). The viral reservoir of

two of the three participants (P6, P7) was previously characterized at T1 by FLIPS and Matched Integration site and Proviral Sequencing (MIP-Seq), providing an extensive resource for comparison with the STIP-Seq assay[21,42]. To this end, NFL proviral genomes obtained with FLIPS and MIP-Seq were also trimmed to the V1–V3 *env* region and included in the phylogenetic trees (Fig. 5).

A total number of 29 p24+ cells at T1 and 17 p24+ cells at T2 were recovered (Fig. 4b). Overall, little differences were observed between the two time points, with most of the clones identified under ART (T1) persisting during the ATI (T2) (Fig. 4b). However, participant P6 displayed a novel clone at T2 when compared to T1, with an IS in the *VMP1* gene (Fig. 4b). Interestingly, the provirus from this clone did not match any V1–V3 *env* SGS, FLIPS, MIP-Seq, or STIP-Seq sequences obtained at T1 (together evaluating $n = 382$ proviruses) (Fig. 5a). In contrast, three out of nine cells recovered by STIP-Seq at T2 yielded this provirus, indicating that this clone emerged or enlarged during the ATI.

For participant P6, one plasma sequence obtained during the ATI (T4) matched a provirus (IS at chr8:10079212) that was recovered with STIP-Seq at T1 ($n = 2$) and T2 ($n = 2$) (Fig. 5a, indicated with a red box). Interestingly, this provirus had a deletion at the 5′-end of the genome covering a large portion of the *p17* gene (Supplementary Fig. 3), making it unlikely that it could produce infectious virions. We previously established that the clonal prediction score (CPS) for the V1–V3 *env* region of participant P6 is 95% (based on $n = 22$ NFL genomes with detectable V1–V3), indicating that while this score is high, this subgenomic region is not capable of differentiating all distinct proviruses[42]. Therefore, we cannot exclude the possibility that this plasma sequence stems from another provirus that has the same V1–V3 *env* sequence, though differs elsewhere in the genome.

For participant P7, five identical plasma sequences recovered at T1 matched an intact provirus (IS in *KCNA3*) that was identified with STIP-Seq at T1 ($n = 4$) and T2 ($n = 3$), indicating that this clone was responsible for low-level viremia (LLV) production under ART (Fig. 5b, indicated with a green box). Interestingly, this clone had a predicted TCR specificity against *M. tuberculosis*, suggesting that clones responsible for LLV on ART can proliferate in response to a circulating antigen (Fig. 4b). Similarly, one plasma sequence recovered during T1 matched a provirus (IS at chr17:7545670) that was identified with STIP-Seq at T1 ($n = 2$) and T2 ($n = 3$) (Fig. 4b). This provirus displayed a 5 bp deletion in stem-loop 2 which removed the MSD, suggesting that a deletion of the MSD would still allow for detectable virion production (Fig. 5b, indicated by a red box). As calculated previously, the CPS for the V1–V3 *env* region of participant P7 is 100% (based on $n = 17$ NFL genomes with detectable V1–V3), giving confidence about the validity of these matches[42]. Interestingly, FLIPS and MIP-Seq identified three additional proviruses that are genome-intact, but were not detected with STIP-Seq (IS in *ZNF274, ZNF141, GGNBP2*) (Fig. 5b). The low number of sampled p24+ cells ($n = 11$ at T1, $n = 8$ at T2, Fig. 4b) could potentially explain this observation, although is it also possible that these proviruses were not induced after a single round of PMA/ionomycin stimulation.

For participant P8, a single clone was identified, with a genome-intact provirus integrated in the *SMG1P2* pseudogene (Fig. 4b). The proviral sequence matched plasma sequences at T1, T2, and T3 ($n = 6$, 3, and 1, respectively), suggesting that this clone was responsible for LLV production under ART, and further contributed to rebound viremia upon ATI (Fig. 5c, indicated by a green box). Because FLIPS data for this participant were not available, the CPS could not be calculated. Alternatively,

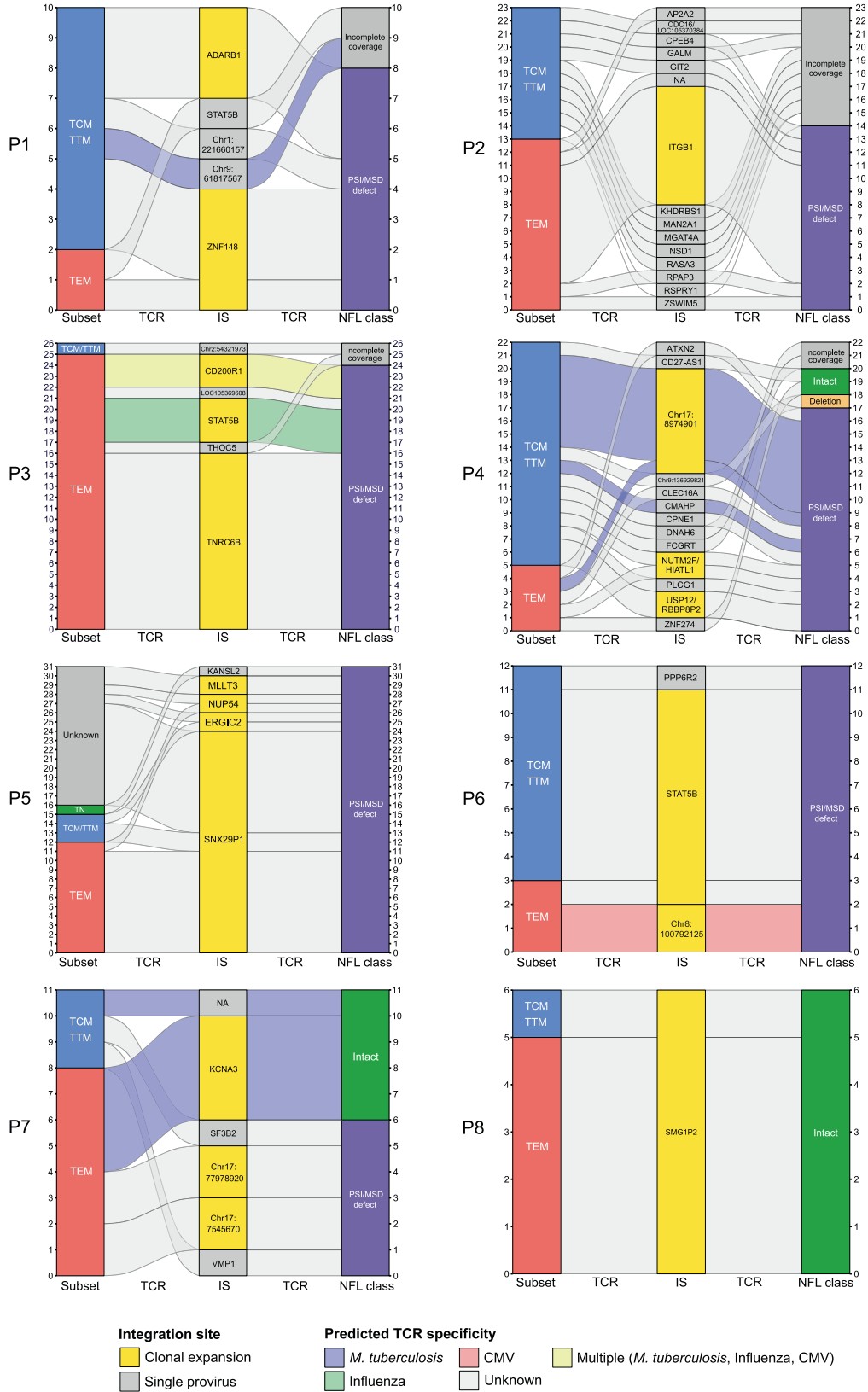

**Fig. 3 Predicted TCR specificity of single p24-producing cells.** Alluvial plots showing the memory phenotype of the host cell, the IS and the NFL class for each p24-producing cell from *n* = 8 ART-treated individuals. Single p24+ sorted cells are represented on the *y*-axis of each plot. Alluvials connecting the different categories are colored according to predicted TCR specificity. Only time points on ART are represented on this figure. IS integration site, TCR T cell receptor, NFL near full-length, TN naïve T cell, TCM central memory T cell, TTM transitional memory T cell, TEM effector memory T cell.

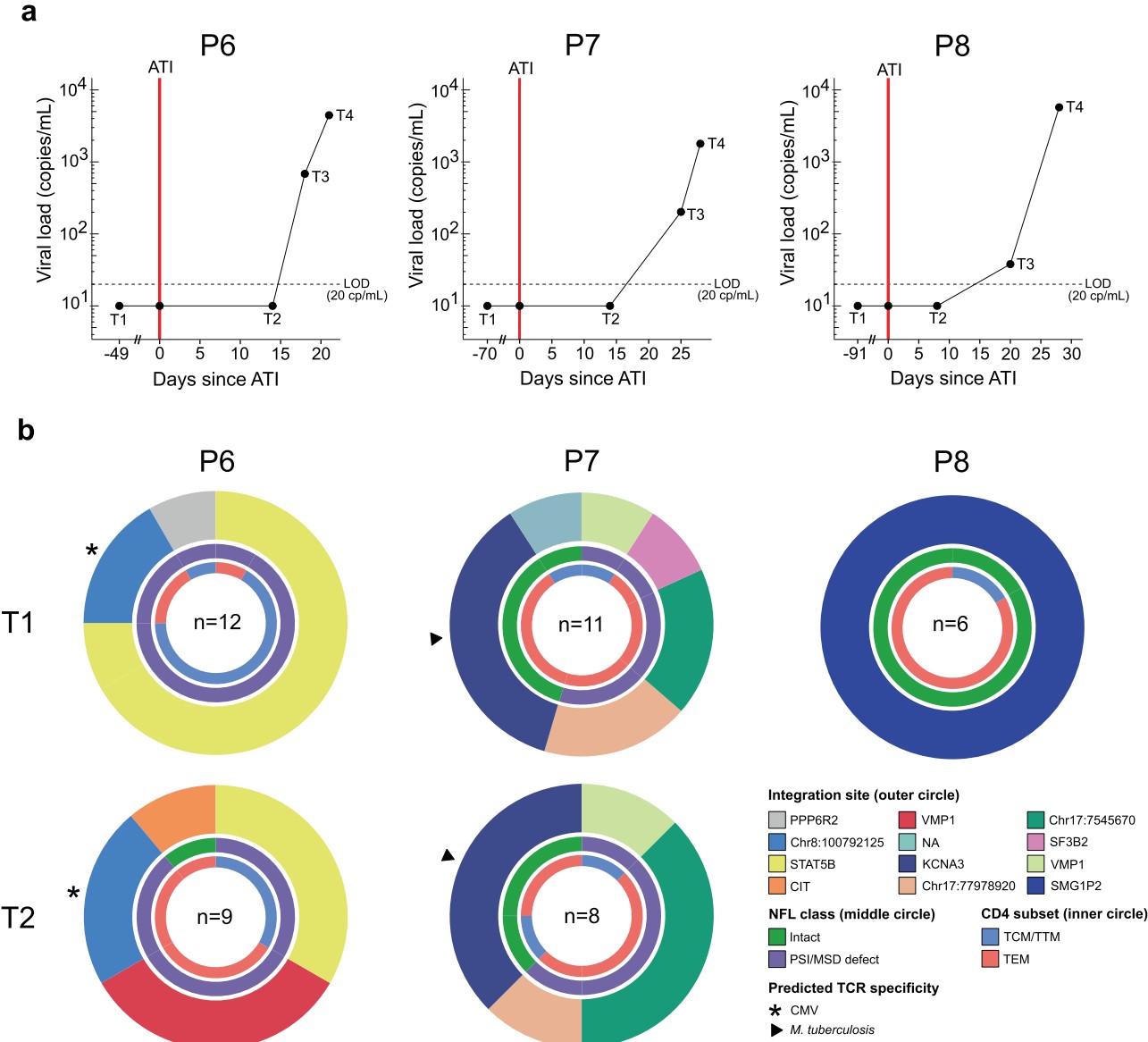

**Fig. 4 STIP-Seq on three participants before and during an analytical treatment interruption (ATI).** **a** Viral load diagram showing the sampling time points before (T1) and during (T2, T3, T4) the analytical treatment interruption for three participants (P6, P7, P8). The viral load was undetectable at T1 and T2, under 1000 cp/mL at T3 (early rebound) and above 1000 cp/mL at T4 (late rebound). The vertical red line depicts the start of the ATI. LOD limit of detection. **b** Donut charts displaying integration sites, NFL class, and memory subsets of p24-producing cells recovered before (T1) and during (T2) an ATI. The number of analyzed p24+ cells is indicated for each participant. PSI packaging signal, MSD major splice donor, TCM central memory T cell, TTM transitional memory T cell, TEM effector memory T cell, NFL near full length.

the nucleotide diversity at T1 was calculated based on proviral V1–V3 *env* sequences, revealing a low diversity (0.00318 vs. 0.01579 for P6 and 0.01805 for P7)[21], which could potentially lead to inaccurate links.

In conclusion, our data suggest that STIP-Seq can capture clones that contribute to LLV and viral rebound, and that in some cases, clones contributing to viral rebound already produce LLV during ART. Furthermore, certain clones responsible for LLV during ART seem to proliferate in response to antigenic stimulation.

## Discussion

HIV cure is impeded by the existence of a persistent viral reservoir, capable of refueling viremia upon treatment interruption. Unraveling mechanisms of viral latency and reservoir maintenance through clonal proliferation are research priorities in the field. Previous studies have shown that reservoir persistence is the result of a complex interplay between proviral genome integrity[22,24], IS[19,26,39], and antigenic stimulation of infected cells[13–15], among other factors. In this regard, several assays have been developed to investigate these factors individually in ART-suppressed individuals. Here, we introduce a novel method to simultaneously characterize the NFL proviral genome and IS of translation-competent proviruses, as well as the phenotype and TCR sequence of the host cells. STIP-Seq requires only a limited amount of CD4 T cells (~5–10 million) and overcomes the need for limiting dilutions, as each sorted p24+ cell is HIV infected. As a result, STIP-Seq is less labor and reagent intensive than MDA-based approaches on bulk DNA.

Conducting STIP-Seq on blood samples from eight ART-suppressed individuals allowed for an in-depth characterization of the translation-competent reservoir. Only 12.5% of proviruses

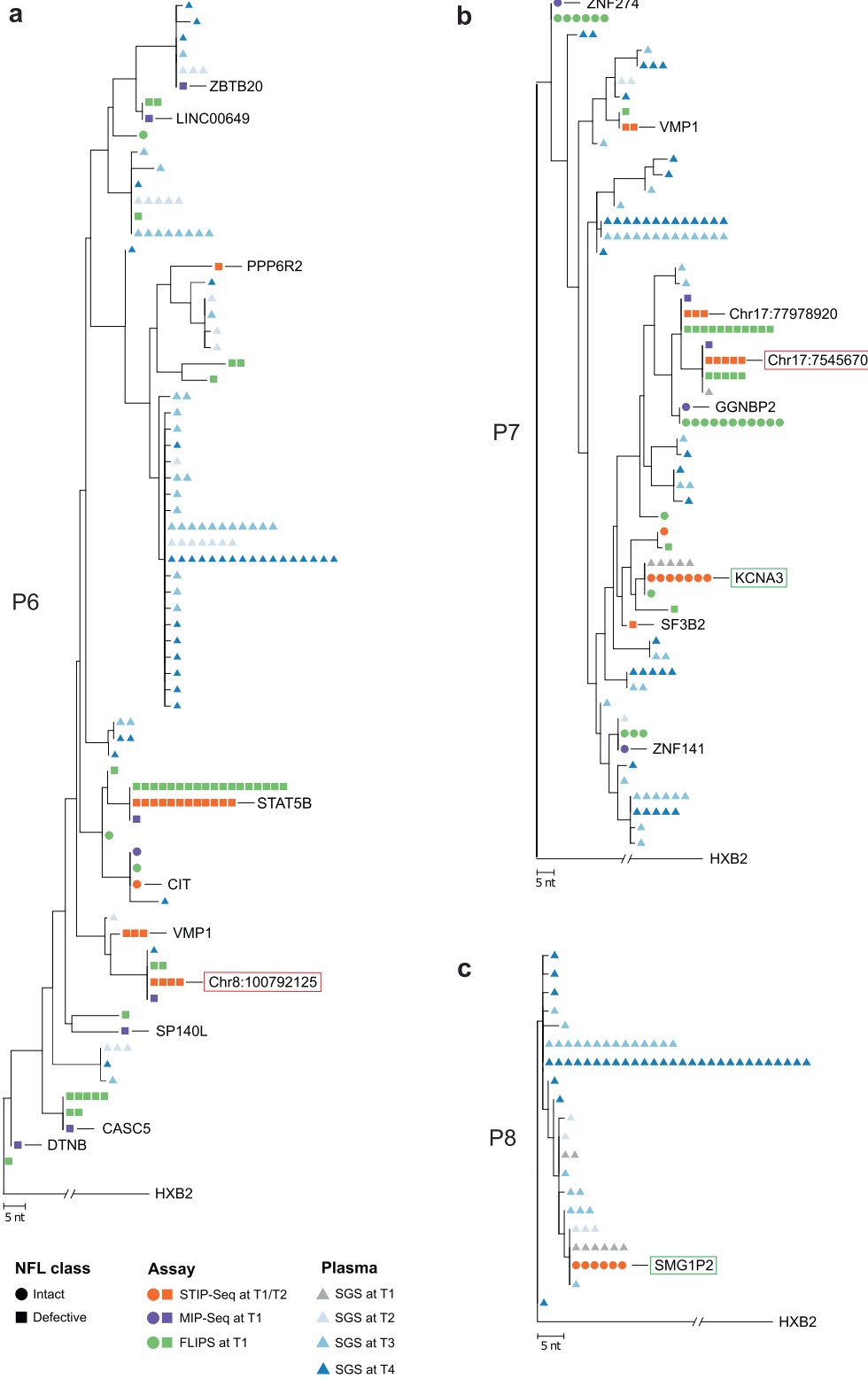

recovered with STIP-Seq were putatively intact, indicating that a large fraction of the translation-competent reservoir might not be replication-competent, as previously suggested[37]. Interestingly, a large proportion (45%) of the proviruses had intact open reading frames for all the protein coding genes, in contrast with proviruses obtained on bulk DNA, which often display large internal deletions, inversions, or hypermutations[22,23]. Nevertheless, most of the proviruses recovered with STIP-Seq had small deletions (<500 bp) at the 5′-end of the genome, frequently involving a deletion of the MSD, as well as the cryptic donor (CD) site located 4 bp downstream of the MSD[24,43]. The presence of either of these sites was previously thought to be essential for correct

**Fig. 5 STIP-Seq identifies clones responsible for viremia under ART and upon treatment interruption. a–c** Maximum-likelihood phylogenetic trees for three participants that underwent an analytical treatment interruption. The trees include V1–V3 *env* plasma sequences from before (T1) and during (T2, T3, and T4) the treatment interruption (P6, P7, and P8), as well as STIP-Seq, MIP-Seq and FLIPS sequences (T1) that were trimmed to the V1–V3 *env* region (P6, P7). Intact and defective proviruses are represented by circles and squares respectively, while V1-V3 plasma sequences are represented by triangles. Each assay is color-coded. Clones displaying a match between defective and intact STIP-Seq sequences and plasma sequences are indicated by red and green frames, respectively. HXB2 subtype B HIV-1 reference genome, NFL near full length, STIP-Seq simultaneous TCR, Integration site and Provirus sequencing, MIP-Seq Matched Integration site and Provirus sequencing, FLIPS Full-Length Individual Provirus Sequencing.

splicing of viral transcripts and subsequent translation into viral proteins[43,44]. However, Pollack et al.[45] showed that proviruses can bypass MSD deletions and mutations by activating alternative splice donor sites. Here, we showed that proviruses with MSD/CD deletions can produce detectable amounts of p24 protein, suggesting that Tat/Rev mRNA can be produced despite MSD/CD mutations and/or that p24 production following PMA/ionomycin stimulation can happen in a Tat/Rev-independent manner. Indeed, an in vitro study showed that Tat-defective HIV-strains can produce readily detectable p24 following PMA stimulation[46]. Also, since it has been reported that PMA increases the expression of active NF-kB and P-TEFb[47], and that NF-kB can directly bind P-TEFb to promote elongation of transcription in a Tat-independent manner[48], it is likely that P-TEFb recruitment by NF-kB increases the levels of unspliced RNA. Furthermore, we showed that a significant proportion of the proviruses recovered with STIP-Seq lacked the AUG start codon of the Gag polyprotein (16/40, 40%), indicating the use of an alternative start codon. We acknowledge the possibility that PMA/ionomycin stimulation could favor the usage of alternative splice sites and/or alternative start codons, which might not reflect the in vivo capacities of the proviruses. Future experiments with alternative latency reversal agents (LRAs) will need to be conducted to investigate whether proviruses with an MSD/CD deletion and/or Gag AUG deletion can splice and translate under physiological conditions.

We also found several translation-competent proviruses with deletions in the packaging signal (PSI), frequently spanning multiple stem-loops. Although this observation suggests that these proviruses are not replication-competent, previous studies have shown that viral genomes can still be packaged despite PSI defects, though with a considerably lower efficiency[45,49]. Importantly, it has been shown that MSD/PSI-defective proviruses can produce viral proteins that can be recognized by cytotoxic CD8 T cells, leading to chronic immune activation[45,50]. We therefore conclude that while STIP-Seq does not solely enrich for genome-intact proviruses, it does enrich for proviruses that are potentially involved in HIV-1 pathogenesis. Future studies on MSD/PSI-defective proviruses will have to be conducted to further elucidate the effect of MSD/PSI deletions on replication capacity, including a detailed assessment of viral splicing products and cloning of MSD/PSI-defective genomes into expression vectors.

We identified three distinct IS into *STAT5B*, a gene that was previously described as a hotspot for HIV-integration in ART-suppressed individuals[16–18]. A previous study has shown that integration in *STAT5B* in the same orientation as the gene can lead to aberrant splicing and subsequent cellular proliferation[18]. Interestingly, the three proviruses identified in the present study were integrated in the reverse orientation. However, studies that reported an overrepresentation of IS in the *STAT5B* gene showed that these IS could be found in both orientations, without an apparent bias[51]. This suggests that integration in the reverse orientation could still be associated with clonal expansion, driven by mechanisms other than virus–host aberrant splicing.

Furthermore, we identified several infected cell clones with predicted specificities towards CMV, *M. tuberculosis,* and influenza, underlining the role of antigen stimulation as a driver of clonal expansion[13–15]. In accordance with results from Simonetti et al., we found clonal cell populations with predicted TCR specificity towards pathogens that harbored proviruses integrated in genes involved in cellular proliferation (*STAT5B*, *KCNA3*), strengthening the hypothesis of a synergetic effect between IS-driven and antigen-driven expansion[15]. Furthermore, our data suggest that one of these clones is responsible for LLV under ART (*KCNA3*), providing evidence that the proliferation of such clones can be driven by antigenic stimulation and/or IS-specific mechanisms.

In the context of an ATI, we compared p24+ cells obtained before (T1) and at the beginning of the ATI (T2). In one participant, a novel clone emerged during the ATI, which was not detected by V1–V3 *env* SGS, FLIPS, MIP-Seq, or STIP-Seq at T1. As we have previously shown that interferon-stimulated genes as well as Tat/Rev transcripts were already upregulated at T2 despite an undetectable viral load[52], we hypothesize that this clonal expansion might have been driven by the inflammatory environment. In addition, we found identical sequences between proviruses recovered with STIP-Seq and plasma viral sequences before and during the ATI, suggesting that HIV-infected clones can produce LLV during ART and/or contribute to rebound viremia upon ATI. This observation is in line with findings from Kearney et al.[53], which showed overlap between proviral p6-PR-RT sequences (DNA and cell-associated RNA, ~1540 bp) and plasma sequences obtained during TI. Because the matching p6-PR-RT sequences were often clonal in nature, this prior study suggested that initial rebound could be fueled by clonally expanded populations of infected cells. Similarly, Aamer et al.[54] identified links between plasma sequences recovered during TI and clonal C2-V5 *env* plasma sequences (~600 bp) that persisted for several years under treatment, supporting the notion that clonal cell populations that produce LLV under ART can contribute to viral rebound. Using MDA-based NFL and ISA on bulk CD4 T cell DNA, Halvas et al.[55] identified clonal populations of proviruses that could be linked to plasma sequences in non-suppressed individuals on ART, though their contribution to rebound viremia was not investigated. In the present study, we provide deeper insights by identifying the phenotype and predicted TCR specificity of such clones and linking them to rebounding plasma sequences.

We acknowledge several limitations to this study. First, due to limited sample availability, we were not able to perform the VOA. Therefore, we could not evaluate the replication-competence of proviral sequences obtained with STIP-Seq by comparing them to sequences from positive VOA wells. Such comparison would have been particularly interesting for the participants that underwent an ATI, given the notoriously poor overlap between sequences derived from VOA and rebound plasma sequences[56–58]. Second, the link to rebound plasma sequences was based on a subgenomic region of the viral genome (V1–V3 *env*). It has previously been shown that using a subgenomic region to link viral sequences is not always adequate, as some viruses share the same subgenomic

sequence while differing elsewhere in the genome[27,59]. However, we previously determined the CPS for two of the three participants that underwent an ATI, revealing high scores: 95% for P6 and 100% for P7 (ref. [42]). Although the CPS should not be considered definitive, these scores give confidence about the validity of the observed matches. Third, TCR specificities of p24+ cells were bio-informatically inferred based on CDR3 sequences, which might not always reflect accurate antigen specificity. However, while clinical data regarding antigen exposure were not available for participants P1 and P3, participant P6 had positive serology for CMV and participant P4 tested positive for the Mantoux tuberculin skin test (*M. tuberculosis*). In contrast, participant P7 tested negative for the Mantoux skin test, despite the detection of a cell clone with predicted specificity towards *M. tuberculosis*. Future work will be needed to confirm the accuracy of CDR3-based TCR predictions. Finally, like other assays based on reactivation of proviruses with an LRA, STIP-Seq probably does not pick up all translation-competent proviruses, as reactivation is a stochastic process[30,47]. In this regard, it has been suggested that the IS can have an influence on the reactivation of the provirus[26,39], and that different LRAs might induce reactivation of distinct proviral species[60]. As such, future studies with STIP-Seq investigating the relationship between different classes of LRAs and the IS of the reactivated proviruses would be of great interest.

In conclusion, our STIP-Seq assay enables deep characterization of the translation-competent HIV reservoir by simultaneously capturing four layers of information: NFL proviral genome, IS, phenotype, and the TCRβ sequence of the host cell. By conducting this assay on ART-suppressed individuals, we provide further insights on the composition of the translation-competent reservoir and its persistence by clonal proliferation. Applying STIP-Seq in the context of an ATI suggested that cell clones harboring translation-competent proviruses contribute to residual viremia and viral rebound upon ART interruption. Using STIP-Seq on a larger cohort of individuals, along with a more elaborate panel of antibodies and different types of LRAs, will help to further unravel the complex interplay between viral and cellular factors involved in the long-term persistence of the HIV reservoir.

## Methods

**Participants and blood collection**. A total of eight individuals on stably suppressive ART were included in this study (Supplementary Table 1). All participants were males, with a mean age of 52.4 years. All participants were infected with HIV-1 subtype B. Participants P1–P4 were recruited at the McGill University Health Centre and the Centre Hospitalier de l'Université de Montréal. Participants P5–P8 were recruited at Ghent University Hospital. Participants P6–P8 are part of the HIV-STAR cohort (Ghent University) (P6 = STAR 10, P7 = STAR 11, P8 = STAR 3). All participants underwent leukapheresis to collect large numbers of PBMCs. PBMCs were isolated by Ficoll density gradient centrifugation and were cryopreserved in liquid nitrogen.

**Ethics statement**. All participants were adults and signed informed consent forms approved by the Ethics Committee of the Ghent University Hospital (Belgium), McGill University Health Centre and Centre Hospitalier de l'Université de Montréal (Canada).

**Antibodies**. Fixable Viability Stain 510 was obtained from ThermoFisher Scientific (#L34957; 1/1000 dilution). The following antibodies were used in sorting experiments: CD8-AF700 Clone RPA-T8 (ThermoFisher, #56-0088-41; 1/200 dilution), CD45RO-BV421 Clone UCHL1 (BD Biosciences, #562649; 1/100 dilution), CD27-BV605 Clone L128 (BD Biosciences, #562656; 1/100 dilution). For p24 staining, we used a combination of two antibodies: p24 KC57-FITC (Beckman Coulter, #6604665; 1/500 dilution) and p24 28B7-APC (MediMabs, #MM-0289-APC; 1/400 dilution).

**Negative selection of CD4 T cells**. CD4 T cells were isolated from peripheral blood mononuclear cells (PBMC) by negative magnetic selection using the EasySep

Human CD4 T Cell Enrichment Kit (StemCell Technology, #19052). Purity was typically >98%.

**HIV-flow procedure**. 5–10 × 10⁶ CD4 T cells were resuspended at $2 \times 10^6$ cells/mL in RPMI + 10% fetal bovine serum and antiretroviral drugs were added to the culture (200 nM raltegravir, 200 nM lamivudine) to avoid new cycles of replication. Cells were stimulated with 1 µg/mL ionomycin (Sigma, #I9657) and 162 nM PMA for 24 h (Sigma, #P8139). Frequencies of p24+ cells were measured by using a combination of two antibodies targeting the p24 protein (p24 KC57-FITC, p24 28B7-APC) as previously described by Pardons et al.[33]. The detailed protocol of the HIV-Flow procedure can be found here: dx.doi.org/10.17504/protocols.io.w4efgte.

**Methanol-based HIV-Flow procedure (STIP-Seq)**. 5–10 × 10⁶ CD4 T cells were resuspended at $2 \times 10^6$ cells/mL in RPMI + 10% fetal bovine serum (FBS; HyClone, #RB35947) and antiretroviral drugs were added to the culture (200 nM raltegravir, 200 nM lamivudine) to avoid new cycles of replication. Cells were stimulated with 1 µg/mL ionomycin (Sigma, #I9657) and 162 nM PMA (Sigma, #P8139). After a 24 h stimulation, a maximum of 10 × 10⁶ cells per condition were resuspended in PBS and stained with fixable viability stain 510 for 20 min at RT. Cells were then stained with antibodies against cell surface molecules (CD8, CD45RO, CD27) in PBS + 2% FBS for 20 min at 4 °C. After a 5-min-centrifugation step at 4 °C to pre-chill the cells, CD4 cells were vortexed to avoid clumping and 1 mL of ice-cold methanol (−20 °C) was gently added. Cells were fixed/permeabilized in methanol for 15 min on ice. Intracellular p24 staining was performed in PBS + 2% FBS using a combination of two antibodies (p24 KC57-FITC, p24 28B7-APC) (45 min, RT). Cells were then washed and resuspended in PBS for subsequent sorting. In all experiments, CD4 T cells from an HIV-negative control were included to set the threshold of positivity. The detailed protocol of the methanol-based HIV-Flow procedure can be found here: https://protocols.io/view/methanol-based-hiv-flow-bpedmja6.

**Single-cell sorting of p24+ cells by FACS**. Single p24+ cells were sorted on a BD FACSAria™ Fusion Cell Sorter. The gating strategy used to sort the cells is represented in Supplementary Fig. 9. Cells were sorted in skirted 96-well PCR plates (Biorad, #12001925) into a volume of 4 µL PBS sc 1× (Qiagen, #150345). To avoid evaporation of the PBS sc 1× during the sort, the PCR plate was continuously chilled at 4 °C. Index sorting was used to enable phenotyping of single-sorted cells. Index sorting is a procedure where coordinates of single-sorted cells for all markers are documented, allowing for retrospective determination of the phenotype of each individual sorted cell. CD4 T cell memory subsets were defined as follows: TN = CD45RO− CD27+, TCM/TTM = CD45RO+CD27+, TEM = CD45RO+CD27−, TTD = CD45RO−CD27− (Supplementary Fig. 9). BD FACSDiva Software v8.0.2 was used to acquire flow cytometry data and Flow-Jo software v10.6.2 was used to analyze flow cytometry data (Tree-Star).

**Multiple displacement amplification**. WGA of single-sorted cells was carried out by MDA with the REPLI-g single cell kit (Qiagen, #150345), according to the manufacturer's instructions. A positive control, consisting of ten p24- cells sorted into the same well, was included on every plate.

**Quantitative polymerase chain reaction (qPCR) for RPP30**. After WGA, reactions were screened by a binary qPCR on the RPP30 reference gene. The PCR mix consisted of 5 µL 2× LightCycler® 480 Probes Master (Roche, #04707494001), 1 µL MDA product, 0.4 µL 10 µM forward and reverse RPP30 primers, 0.2 µL 10 µM RPP30 probe, and 3 µL nuclease free water. Cycling was performed on a LightCycler 480 machine (Roche) according to the following program: 10 min at 95 °C; 45 cycles (10 s at 95 °C, 1 min at 56 °C, 1 s at 72 °C); 10 s at 40 °C. Reactions that yielded a cycle of threshold (Ct) value of 38 or lower, were selected for further downstream processing (Supplementary Fig. 2). The primer and probe sequences are given in Supplementary Fig. 5.

**IS analysis**. MDA reactions that were positive for RPP30 were subjected to a modified version of the ISLA assay, capturing the integration site either upstream (5′) or downstream (3′) of the integrated provirus[16,42]. Linear extension was performed on 1 µL MDA product, using either the up3.2 (3′) or UTR.629.R (5′) primer. The reaction consisted of 5 µL Advantage2 10× buffer and 0.5 µL Advantage2 polymerase (Takara Bio, #639202), 0.2 µL 10 mM dNTP, 0.75 µL 20 µM primer, and 42.55 µL nuclease free water. Cycling was performed as follows: 2 min at 95 °C; 30 cycles (20 s at 94 °C, 1 min at 60 °C, 1 min 15 s at 72°C). Next, annealing and extension of random-decamer looping primers was performed using the deca1.U5 (3′) or decaU3R.3 (5′) primer. The reaction consisted of 50 µL previous round product, 8 µL 20 µM primer, 2 µL Mytaq DNA polymerase (Bioline, # BIO-21105), and 20 µL nuclease free water. Cycling was performed as follows: 2 min at 68 °C; 1 min at 65 °C; cooled by 1 °C/min until reaching 25 °C; 1 min at 60 °C; cooled by 1 °C per minute until reaching 20 °C. Formation of loops and first-round PCR amplification were performed with the RF2 (3′) or U3R.1 (5′) primer. The reaction consisted of 20 µL previous round product, 7 µL Mytaq buffer, and 0.75 µL Mytaq DNA polymerase (Bioline, # BIO-21105), 1.5 µL 20 µM primer, and

20.75 μL nuclease free water. Cycling was performed as follows: 2 min at 95 °C; 10 cycles (20 s at 94 °C, 30 s at 60 °C, 2 min at 72 °C); 40 cycles (10 s at 92 °C, 15 s at 65 °C, 2 min at 72 °C); 5 min at 72 °C. Second and third round nested PCR amplification were performed with the RF1/1.U5 (3′) or U3R.2/U3R.3 (5′) primers. The reaction consisted of 2 μL previous round product, 5 μL Mytaq buffer and 0.375 μL Mytaq DNA polymerase (Bioline, # BIO-21105), 0.75 μL 20 μM primer, and 16.375 μL nuclease-free water. Cycling was the same for both rounds, as follows: 2 min at 95 °C; 35 cycles (20 s at 94 °C, 30 s at 65 °C, 2 min at 72 °C); 5 min at 72 °C. Resulting amplicons were visualized on a 1% agarose gel and Sanger sequenced using the 2.U5 (3′) or U3R.4 (5′) primers (Eurofins Genomics). Analysis of the sequences was performed using the "IntegrationSites" webtool (https://indra.mullins.microbiol.washington.edu/integrationsites). Cancer-related genes were identified based on the "allOnco" gene list, available online: http://www.bushmanlab.org/links/genelists. All ISLA primers are summarized in Supplementary Table 5.

**Near full-length proviral genome amplification.** Near full-length HIV-1 proviral sequencing was performed on MDA wells that were RPP30 positive (Supplementary Fig. 1). First, a set of five non-multiplexed nested PCRs was used to amplify the proviral genome, yielding five amplicons of approximately 2 kb in length that together cover 92% of the HIV-1 genome, as described[26,61]. Briefly, 1 μL of 1/5 diluted MDA product was added to 2.5 μL Platinum Taq High Fidelity buffer and 0.125 μL Platinum Taq High Fidelity polymerase (Invitrogen, #11304011), 1 μL 50 mM MgSO₄, 0.5 μL 10 mM dNTP, 0.25 μL 100 μM forward and reverse primers, and 19.375 μL nuclease-free water. The second-round PCR had the same composition, but took 1 μL first-round product as input. Cycling was the same for both rounds, as follows: 2 min at 94 °C; 35 cycles (15 s at 94 °C, 30 s at 55 °C, 2 min 30 s at 68 °C); 5 min at 68 °C.

MDA wells that did not yield an amplicon for all five PCRs were subjected to left and right half genome amplification. The 25 μL PCR mix for the first round is composed of 5 μL 5× Prime STAR GXL buffer, 0.5 μL PrimeStar GXL polymerase (Takara Bio, #R050B), 0.125 μL ThermaStop (Sigma Aldrich, #TSTOP-500), 250 nM forward and reverse primers, and 1 μL MDA product. The mix for the second round has the same composition and takes 1 μL of the first-round product as an input. Thermocycling conditions for first and second PCR rounds are as follows: 2 min at 98 °C; 35 cycles (10 s at 98 °C, 15 s at 62 °C, 5 min at 68 °C); 7 min at 68 °C.

For selected wells, NFL amplification using a set of four non-multiplexed PCRs was performed. Briefly, 1 μL MDA product was added to 4 μL Platinum Taq High Fidelity buffer and 0.2 μL Platinum Taq High Fidelity polymerase (Invitrogen, #11304011), 1.6 μL 50 mM MgSO₄, 0.8 μL 10 mM dNTP, 0.4 μL 100 μM forward and reverse primers, and 31.6 μL nuclease-free water. Cycling was performed as follows: 2 min at 94 °C; three cycles (30 s at 94 °C, 30 s at 64 °C, 10 min at 68 °C); 3 cycles (30 s at 94 °C, 30 s at 61 °C, 10 min at 68 °C); three cycles (30 s at 94 °C, 30 s at 58 °C, 10 min at 68 °C); 21 cycles (30 s at 94 °C, 30 s at 55 °C, 10 min at 68 °C); 10 min at 68 °C. The second-round reaction consisted of 1 μL first-round product, 2.5 μL Platinum Taq High Fidelity buffer and 0.125 μL Platinum Taq High Fidelity polymerase (Invitrogen, #11304011), 1 μL 50 mM MgSO₄, 0.5 μL 10 mM dNTP, 0.25 μL 100 μM forward and reverse primers, and 19.375 μL nuclease-free water. The cycling conditions were identical to those of the 5-amplicon approach.

FLIPS was performed on DNA extracted from total CD4 T cells with the DNeasy Blood & Tissue Kit (Qiagen, #69504), as described[22]. Briefly, the first-round PCR was identical to the first round of the 4-amplicon approach. The second-round reaction consisted of 2 μL 1:3 diluted first round product, 3 μL Platinum Taq High Fidelity buffer and 0.2 μL Platinum Taq High Fidelity polymerase (Invitrogen, #11304011), 1.2 μL 50 mM MgSO₄, 0.6 μL 10 mM dNTP, 0.3 μL 100 μM forward and reverse primers, and 22.5 μL nuclease-free water. The cycling conditions were identical to those of the first round PCR.

Amplicons were visualized on a 1% agarose gel and positives were selected for subsequent sequencing. The primer sequences for the five-, two-, four-amplicon, and FLIPS approaches are summarized in Supplementary Table 5.

**Near full-length proviral genome sequencing and assembly.** Amplicons were pooled at equimolar ratios and cleaned by magnetic bead purification (Ampure XP, Beckman Coulter, #A63881), followed by quantification with the Quant-iT Pico-Green dsDNA Assay Kit (Invitrogen, #P7589). Library preparation was done with the Nextera XT DNA Library Preparation Kit (Illumina, Cat. No FC-131-1096) according to the manufacturer's instructions, with indexing of 96-samples per run. The library was sequenced on a MiSeq Illumina platform via a 2 × 150 nt paired-end sequencing with the 300 cycle v2 kit (Illumina, #MS-102-2002) according to the manufacturer's instructions, yielding approximately 200,000 reads per sample. Near full-length proviral genome sequences were de novo assembled as follows: (1) FASTQ quality checks were performed with FastQC v0.11.7 (http://www.bioinformatics.babraham.ac.uk/projects/fastqc) and removal of Illumina adaptor sequences and quality-trimming of 5′ and 3′ terminal ends was performed with bbmap v37.99 (sourceforge.net/projects/bbmap/). (2) Trimmed reads were de novo assembled using MEGAHIT v1.2.9 (ref. [62]) with standard settings. (3) Resulting contigs were aligned against the HXB2 HIV-1 reference genome using blastn v2.7.1 (ref. [63]) with standard settings, and contigs that matched HXB2 were retained. (4) Trimmed reads were mapped against the de novo assembled HIV-1 contigs to generate final consensus sequences based on per-base majority consensus calling,

using bbmap v37.99 (sourceforge.net/projects/bbmap/). Scripts concerning de novo assembly of HIV-1 genomes can be found at the following GitHub page: https://github.com/laulambr/virus_assembly[64]. Consensus proviral genomes had a mean sequencing depth of 3767× with 95% confidence interval (3571, 3964) per-base position. A minimum per-base coverage of 1000× was set as a threshold to ensure accurate results. The sequencing error within our dataset was estimated using viral genomes with a shared IS as a proxy. Performing mean pairwise distance calculations, an average per-base error rate of 0.0019% was obtained, which is equivalent to 1 nt error per 52,631 nucleotides sequenced (with 95% CI 1 per 132,857 to 1 per 32,461).

**Proviral genome classification.** NFL proviral genome classification was performed using the publicly available "Gene Cutter" and "Hypermut" webtools from the Los Alamos National Laboratory HIV sequence database (https://www.hiv.lanl.gov). Proviral genomes were classified in the following sequential order: (1) "Inversion": presence of internal sequence inversion, defined as region of reverse complementarity. (2) "Large internal deletion": internal sequence deletion of >1000 bp. (3) "Hypermutated": APOBEC-3G/3F-induced hypermutation. (4) "PSI/MSD defect": deletion >7 bp covering (part of) the packaging signal region, or absence of GT dinucleotide at the MSD and GT dinucleotide at the cryptic donor site (located 4 bp downstream of MSD)[24]. Proviruses with a deletion covering PSI/MSD that extended into the *gag* gene—thereby removing the *gag* AUG start codon —were also classified into this category. (5) "Premature stop-codon/frameshift": premature stop-codon or frameshift caused by mutation and/or sequence insertion/deletion in the essential genes *gag, pol* or *env*. Proviruses with insertion/deletion >49 nt in *gag*, insertion/deletion >49 nt in *pol*, or insertion/deletion >99 nt in *env* were also classified into this category. (6) "Intact": proviruses that displayed none of the above defects were classified into this category.

**Plasma virus single-genome sequencing.** Sequencing of the V1–V3 *env* region of plasma virus was performed previously by a single-genome sequencing assay with single-copy sensitivity, as described[21,65]. Plasma samples with detectable or undetectable viral load were diluted to 7 mL with Tris-buffered saline and subjected to ultracentrifugation at 170,000 × *g* for 30 min at 4 °C[65]. Viral RNA extraction and cDNA generation was performed as follows: (1) 100 μL 5 mM Tris-HCl (pH 8.0) and 200 μg proteinase K were added, followed by incubation for 30 min at 55 °C. (2) 400 μL 5.8 M guanidinium isothiocyanate and 200 μg glycogen were added. (3) 500 μL of 100% isopropanol alcohol was added, followed by a 15 min centrifugation at 21,000 × *g*, at room temperature. (4) The supernatant was removed and the pellet was washed twice with 70% ethanol. (5) cDNA was generated with SuperScript III Reverse Transcriptase (ThermoFischer, Cat. No 18080093) and the E115 primer (5′-AGAAAAATTCCCCTCCACAATTAA-3′). cDNA was diluted to endpoint to ensure the presence of single HIV-1 copies per reaction, and nested V1–V3 *env* PCR was performed as described[21] (at the dilution where <30% of reactions were positive). Reactions without reverse transcriptase were negative, ensuring that the RNA extracts were not contaminated by DNA. PCR products were checked on a 1% agarose gel and positives were sequenced by Sanger sequencing (Australian Genome Research Facility, Sydney, Australia).

**Phylogenetic analyses.** Sequences obtained with STIP-Seq, MIP-Seq, and FLIPS were trimmed to the V1–V3 *env* region and multiple aligned to V1–V3 *env* sequences from plasma using MAFFT[66]. Phylogenetic trees were constructed using PhyML v3.0 (best of NNI and SPR rearrangements) and 1000 bootstraps[67].

**TCR sequencing.** A previously developed two-step PCR method to amplify a portion of approximately 260 bp of TCRβ (including the CDR3 region) was applied to MDA-positive wells[14]. Briefly, a multiplex PCR was performed using a set of 35 primers tailed with M13 forward and reverse priming sites using the Qiagen Multiplex PCR Kit (Qiagen, #206143). The reaction consisted of 1 μL MDA product, 25 μL of Qiagen Multiplex PCR 2× master mix, 10 μL of primer mix (250 nM per primer), 5 μL Q solution and 9 μL nuclease-free water. Cycling was performed as follows: 15 min at 95 °C; 40 cycles (30 s at 95 °C, 90 s at 68 °C, 20 s at 72 °C); 5 min at 72 °C. The second-round reaction consisted of 10 μL first round product, 5 μL Taq DNA Polymerase buffer and 0.5 μL Taq DNA Polymerase (Invitrogen, #10342053), 3 μL 50 mM MgCl₂, 1.5 μL 10 μM M13F primer, 1.5 μL 10 μM M13R primer, and 26 μL nuclease-free water. Cycling was performed as follows: 15 min at 95 °C; 40 cycles (30 s at 95 °C, 90 s at 57 °C, 30 s at 72 °C); 5 min at 72°C. Amplicons were visualized on a 1% agarose gel and positives were Sanger sequenced with the M13F and M13R primers. TCRβ sequences were reconstructed using both forward and reverse sequences, and were analyzed using the V-QUEST tool of the IMGT® database (IMGT®, the international ImMunoGeneTics information system® [http://www.imgt.org]). All primers for TCR sequencing are summarized in Supplementary Table 5.

**Prediction of TCR specificity.** TCR sequences were analyzed using an algorithm to predict antigen specificity: CDR3 sequences were compared to the McPAS-TCR database of TCRs of known antigenic specificity (http://friedmanlab.weizmann.ac.il/McPAS-TCR/)[68] and sequence similarities were identified. We predicted TCR specificity using the three criteria described by Meysman et al.[69]: (1) Query CDR3 sequences should be at least nine amino acids long; (2) query and hit

CDR3 sequences should have identical length; (3) query and hit CDR3 sequences should not differ by more than one amino acid (Levenshtein distance ≤1). Among all CDR3 sequences, those fulfilling these three criteria with matched CDR3 sequences from the database were considered at high probability of sharing the same specificity.

**Data representations and statistical analyses**. Bar charts, line plots, and donut plots were generated in R (version 3.4.3) or Graphpad Prism (version 8.0.2). Alluvial plots were generated in R (version 3.4.3) using the ggalluvial package (version 0.12.3) and the ggplot2 package (version 3.3.3). MEGA7 and iTOL v5 were used to visualize phylogenetic trees[70,71]. For correlation testing, the Spearman's rank correlation coefficient was calculated. To compare p24 antibody fluorescence intensities (Supplementary Fig. 4), a $z$-score (standard score) normalization was performed. For group comparisons, two-sided non-parametric Wilcoxon signed-rank tests were used. For comparison of $2 \times 2$ and $2 \times 3$ contingency tables, two-sided Fisher exact testing was used. $P$ values of less than or equal to 0.05 were considered statistically significant.

**Reporting summary**. Further information on research design is available in the Nature Research Reporting Summary linked to this article.

## Data availability

HIV-1 sequence data that support the findings of this study have been deposited in Genbank with the accession codes MW881651–MW881770 and MH642355–MH643573. External databases used in this study are available online: IMGT® database (IMGT®, the international ImMunoGeneTics information system®, http://www.imgt.org); McPAS-TCR database (http://friedmanlab.weizmann.ac.il/McPAS-TCR/); Los Alamos HIV Sequence database (https://www.hiv.lanl.gov); Integration Sites webtool (https://indra.mullins.microbiol.washington.edu/integrationsites); allOnco gene list (http://www.bushmanlab.org/links/genelists). All other data generated or analyzed during this study are included in this paper and its Supplementary Information files. Data underlying main and supplementary figures are provided with this paper as a Source Data file. Source data are provided with this paper.

## Code availability

Scripts concerning de novo assembly of HIV-1 genomes are freely available on GitHub: https://github.com/laulambr/virus_assembly[64]. A description of the key operations and instructions on how to install and run the code are provided on the GitHub page. In addition, a test dataset and a description of expected outputs with expected run times are provided.

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

## Acknowledgements

We thank all participants who donated samples to the study, as well as MDs and study nurses who helped with the recruitment and coordination of this study and the processing of blood samples. The study team thanks Sophie Vermaut for assisting with the flow cytometry platform. We also thank Bram Parton, Céline Helsmoortel, and Kim De Leeneer for helping with the Illumina sequencing. We are grateful for the interesting scientific input and technical help given by Rémi Fromentin, Caroline Dufour, Amélie Pagliuzza, and Sofie Rutsaert. In addition, we thank Jean-Pierre Routy and Josée Girouard for the recruitment of the participants in Montreal. This current research work was supported by the NIH (R01-AI134419, MPI: L.V. and J.I.M. and partially by R01-AI152979) and the Research Foundation Flanders (S000319N and G0B3820N). This work was partially supported by the Canadian Institutes for Health Research (CIHR; operating grant #364408 and the Canadian HIV Cure Enterprise (CanCURE) Team Grant HB2-164064). B.C. was supported by FWO Vlaanderen (1S28918N). L.L. was supported by FWO Vlaanderen (1S29220N). L.V. was supported by the Research Foundation Flanders (1.8.020.09.N.00) and the Collen-Francqui Research Professor Mandate. M.P. was supported by postdoctoral funding from VLAIO O&O (HBC.2018.2278). P.G. was supported by a postdoctoral fellowship from CIHR (#415209), and N.C. was supported by Research Scholar Career Awards of the FRQ-S (#253292). S.P. was supported by the Delaney AIDS Research Enterprise (DARE) to Find a Cure (1U19AI096109 and 1UM1AI126611-01) and the Australian National Health and Medical Research Council (APP1061681 and APP1149990).

## Author contributions

B.C., M.P., L.L., W.W., N.C., and L.V. conceptualized the experiments. Additional scientific input was given by N.C., S.P., W.W., and L.V. N.C. and L.V. provided the samples used in the study. B.C., M.P., L.L., Y.N., and N.B. performed experiments involving cell sorting, multiple displacement amplification, single NFL proviral sequencing, and integration site sequencing. S.P. provided protocols and resources for FLIPS sequencing. P.G. performed TCR sequencing. J.I.M. and L.C. provided protocols to perform the 2-amplicon PCR for NFL proviral sequencing. B.C. and M.P. wrote the paper. All authors read and edited the paper.

## Competing interests

The authors declare no competing interests.
