## [Peer Review File · Nature Communications]

Reviewers' Comments:

Reviewer #1:

Remarks to the Author:

The manuscript by Cole et al. brings together several tools for the analysis of the latent HIV reservoir to provide a new view. The authors use expression of p24 after induction of latent reservoir cells as the basis of sorting cells for the analysis of DNA. Not surprisingly, the subset of proviruses that can express p24 is different from the total HIV DNA reservoir, although it is nice to see the specifics of those differences. They are able to define clones, integration sites, and make predictions about TCR specificity, all possible because of single cell sorting of p24+ cells followed by MDA to provide plenty of DNA to work with for each cell. A big contribution of this paper is to show that they can get good permeabilization and p24 staining using methanol, leaving the DNA intact for subsequent amplification. This is in essence a methods paper but with the emphasis on the plural of methods as the authors bring together a string of high level and topical methods to address fundamental questions. I enjoyed reading this paper both for its content and for the clarity of the writing. I suspect this paper will be handed to graduate students in a number of labs with the comment "We have to get this set up in our lab." I congratulate the authors on putting together what appears to be a series of long distance collaborations to create something where the sum is greater than the parts. I have only a few minor comments/questions.

1. The authors refer to the participants being exposed to certain pathogens (CMV, Tb, Flu). What is the evidence for these exposures? Are these exposures presumed? Is there antibody evidence? If these are assumptions based on the TCRs it seems stronger for CMV and Flu. However, there are populations where Tb exposure is pretty low thus it is not obvious that everyone in this cohort would necessarily have Tb antigen exposure. TCR specificity is an important question in the biology of the reservoir so it would be nice to know what is known and what is guesswork. In this regard it seems odd that the same TCR would target two different pathogens making this seem more like a measure of our lack of clear predictions. Also, two rules cited seem contradictory; on line 545 rule 1 is identical length and rule 2 is length long enough (for what?). If these are not contradictory then a little more explanation would help.
2. I would have been interested in seeing two topics discussed a bit more. Perhaps there is a word limit but I think these points deserve a sentence or two. First, did the clones favor one type of provirus structure? There may not be enough data but I think it is worth a look. Second, the discussion of ATI misses an important point. Most of the text is spent discussing T1 and T2, where T2 is after ATI but before increased viremia. The discussion of LLV is fine but the meaning of T2 is ambiguous – is it ATI for more LLV? I would prefer more discussion of T3 where there is clear viremia. There is always a discussion in the field of whether it is possible to find the rebound virus in the reservoir. If there is a single cell that gives rise to rebound then it is not possible to both observe that cell in the lab and have it give rise to rebound. However, if rebound is from a clone of cells then this is possible. A shift from the data at T2 to T3 would give the authors a more interesting point to discuss.
3. What fraction of the cell clones had integration sites within genes that have been implicated in being able to contribute to cell proliferation? Integration into such genes is well documented but my sense is that it happens only in a small minority of clones yet the authors spend a lot of time on this point. Does the STIP protocol select for these types of integration sites (at least inferred if they are over-represented)?
4. Please give the reader a little more context for "index sorting" when it first appears in the Results.
5. The deletions in p17 are curious. Might there be a bias in detecting these proviruses? The loss of p17/MA/membrane targeting may result in an accumulation of p24/CA/Gag in the cytoplasm making this a robust signal.

Reviewer #2:

Remarks to the Author:

In this study, Cole et al. described a novel single-cell technique, STIP-seq, which allows simultaneous detection of HIV integration sites, proviral genotypes, TCR sequences, and phenotypes of p24-translation-competent HIV-infected cells from clinical samples. Using STIP-seq, the authors examined eight HIV-infected individuals, obtained 158 p24+ cells which yielded 156 integration sites, 40 distinct proviral genomes, and 43 distinct TCR-beta CDR3 sequences. The authors observed that (1) p24+ cells preferentially displayed a memory phenotype and were enriched in proviral genomes with 5' deletions in particular in the MSD stem-loop 2 region, (2) some p24+ cells were associated with TCR sequences specific to *M. tuberculosis*, influenza and CMV and which had undergone clonal expansion suggested by integration site analyses, and finally (3) plasma viral envelope sequences derived from time points immediately before and after virologic rebound shared sequence identity with STIP-seq-derived sequences and provided evidence that clonally-expanded translation-competent infected cells could have contributed to low level viremia and/or plasma virologic rebound.

Overall, this is a very well-written manuscript and a very well-designed study. Techniques and analyses are sophisticated and robust. Conclusions are valid. STIP-seq is an important advancement in the field of HIV persistence research and enables high-resolution interrogation of the persistent viral reservoir. Perhaps the most important distinction from previous reports and technologies is that STIP-seq enables the authors to link antigen-specificity to viral genome-intactness and translation-competency of the proviral genomes.

Major comments:

1. I am surprised to see that only five out of the 40 distinct proviral genomes detected were genome-intact and many had no Gag ATG and still produced detectable p24. Could this be attributable to PMA/i stimulation leading to abnormal start codon usage? What was the MFI of the Gag+ populations? Did cells that have Gag ATG displayed higher fluorescence?
2. Sequencing accuracy is key to genome-intactness versus defectiveness inferences. Please report metrics such as min/max/median sequencing depth per base position and give more details to the de novo assembly step. Were the viral genomes that shared identical integration sites 100% sequence-identical? That will give an estimate of sequencing error.
3. Due to the short V1-V3 env sequence length from the plasma compartment, it is not absolute that the clones identified from STIP-seq are responsible for LLV and/or rebound (as the authors have acknowledged in text). I do agree that it is likely. I would suggest avoiding confirmative language such as "STIP-Seq captures clones that contribute to LLV and viral rebound" (line 274, also line 358, 408, abstract etc).

Minor comments:

1. Line 109. Only 2- and 5- was mentioned. Was the 4-amplicon approach used?
2. The original design of the 5-amplicon primer set was actually developed by a Vancouver team Lee GQ AIDS 2017 (PMID: 28832407) for pan-subtype NFL plasma HIV sequencing, and was later validated and applied to MIP-seq Einkauf KB JCI 2019 (PMID: 30688658). Please consider citing in Supp Table 5.
3. Line 181-182. Did you see identical TCR but non-identical integration sites?
4. Line 195. Since antigen specificities were bioinformatically-inferred, these cells were not technically "antigen-responsive"
5. Line 222. Plasma viral sequencing was not in the methods section. If both T1 and T2 had undetectable viral load, did you use an ultrasensitive approach? Please add to methods section.
6. Line 350. Instead of "could still lead to", consider "could still be associated with"

Reviewer: Guinevere Q. Lee

REVIEWER COMMENTS

Reviewer #1 (Remarks to the Author):

The manuscript by Cole et al. brings together several tools for the analysis of the latent HIV reservoir to provide a new view. The authors use expression of p24 after induction of latent reservoir cells as the basis of sorting cells for the analysis of DNA. Not surprisingly, the subset of proviruses that can express p24 is different from the total HIV DNA reservoir, although it is nice to see the specifics of those differences. They are able to define clones, integration sites, and make predictions about TCR specificity, all possible because of single cell sorting of p24+ cells followed by MDA to provide plenty of DNA to work with for each cell. A big contribution of this paper is to show that they can get good permeabilization and p24 staining using methanol, leaving the DNA intact for subsequent amplification. This is in essence a methods paper but with the emphasis on the plural of methods as the authors bring together a string of high level and topical methods to address fundamental questions. I enjoyed reading this paper both for its content and for the clarity of the writing. I suspect this paper will be handed to graduate students in a number of labs with the comment "We have to get this set up in our lab." I congratulate the authors on putting together what appears to be a series of long distance collaborations to create something where the sum is greater than the parts. I have only a few minor comments/questions.

We thank the reviewer for the positive evaluation of our work. Please find our point-by-point response to the remarks below.

1. The authors refer to the participants being exposed to certain pathogens (CMV, Tb, Flu). What is the evidence for these exposures? Are these exposures presumed? Is there antibody evidence? If these are assumptions based on the TCRs it seems stronger for CMV and Flu. However, there are populations where Tb exposure is pretty low thus it is not obvious that everyone in this cohort would necessarily have Tb antigen exposure. TCR specificity is an important question in the biology of the reservoir so it would be nice to know what is known and what is guesswork.

We agree with the reviewer that it is important to mention evidence for antigen exposure, as we acknowledge that CDR3-based predictions might not always be accurate. The predicted antigen specificities would require functional validation by exposing cells from these participants to the putative antigens followed by confirmation of the CDR3 sequence. Although we are currently pursuing this approach, we feel these experiments are beyond the scope of the current manuscript. However, following the recommendation of the reviewer, we examined pre-existing clinical data from these participants. Interestingly, we found that participant P6 had positive serology for CMV and participant P4 tested positive for the Mantoux tuberculin (PPD) skin test, supporting our predictions from TCR sequences. In contrast, participant P7, from whom some p24+ cells were predicted to be specific to mTB, tested negative for the Mantoux tuberculin (PPD) skin test. Pre-existing clinical data were not available for participants P1 and P3 and due to ethical committee restrictions, these

participants could not be tested at present. We have now discussed these points in the 'limitations' paragraph of the discussion (lines 408-415).

In this regard it seems odd that the same TCR would target two different pathogens making this seem more like a measure of our lack of clear predictions.

We acknowledge that CDR3-based predictions might not always be accurate. However, cross-reactive TCRs have previously been reported (Sewell, *Nat Rev Immunol*, 2012; Antunes *et al.*, *Front Immunol*, 2017; Moorlag *et al.*, *Clin Microbiol Infect*, 2019). Although functional validation would be needed to ascertain the antigen specificity of these cells, we cannot exclude the possibility that the clone with the predicted cross-reactive TCR in participant P3 indeed recognizes epitopes from different pathogens.

Also, two rules cited seem contradictory; on line 545 rule 1 is identical length and rule 2 is length long enough (for what?). If these are not contradictory then a little more explanation would help.

We thank the reviewer for pointing this out. We have now clarified that the length of the CDR3 sequence must be 9 amino acids at minimum. These changes can be found at lines 618-620.

2. I would have been interested in seeing two topics discussed a bit more. Perhaps there is a word limit but I think these points deserve a sentence or two. First, did the clones favor one type of provirus structure? There may not be enough data but I think it is worth a look.

As the reviewer suggested, we compared the NFL class proportions between unique proviruses and proviruses stemming from cell clones. There was no significant difference between "unique" and "clonal" proviruses in terms of NFL class distribution ($p = 0.99$, two-tailed Fisher's exact test). We have now included the figure below in the manuscript as Supplementary Fig. 6, described in the main text at lines 156-160.

Second, the discussion of ATI misses an important point. Most of the text is spent discussing T1 and T2, where T2 is after ATI but before increased viremia. The discussion LLV is fine but the meaning of T2 is ambiguous – is it ATI for more LLV? I would prefer more discussion of T3 where there is clear viremia. it is not possible to both observe that cell in the lab and have it

give rise to rebound. There is always a discussion in the field of whether it is possible to find the rebound virus in the reservoir. If there is a single cell that gives rise to rebound then it is not possible to both observe that cell in the lab and have it give rise to rebound. However, if rebound is from a clone of cells then this is possible. A shift from the data at T2 to T3 would give the authors a more interesting point to discuss.

We agree with the statement that identical links between cellular proviruses and plasma viruses can only be made when the provirus originates from a clonal cell population. In this regard, we and others have previously shown that a large fraction of rebound virus seems to have a clonal origin, increasing the chances of identifying links (De Scheerder *et al.*, *Cell Host Microbe*, 2019; Kearney *et al.*, *J Virol*, 2015; Aamer *et al.*, *PLoS Pathog*, 2020).

We understand the reviewer's point regarding a switch from T2 to T3. However, because of limited sample availability (leukapheresis was not performed at T3), we were not able to perform STIP-Seq at this time point. In addition, when performing STIP-Seq in a setting of detectable viremia (T3/T4), the p24+ fraction is likely to be dominated by productively infected CD4 T cells, rather than reactivated reservoir cells. As our goal was to find the cellular origin of rebound virus, this might have proven difficult using T3/T4 samples. Moreover, we previously showed that interferon-stimulated genes, as well as Tat/Rev transcripts, were significantly upregulated at T2 compared to T1, suggesting the existence of a viral response at that time point despite a VL below the limit of detection of 20 copies/mL (De Scheerder *et al.*, *J Antimicrob Chemother*, 2020; described in the discussion part lines 375-377). Therefore, we hypothesized that the inflammatory environment present at T2 might have favored the reactivation of proviruses responsible for viral rebound. We have now added a description of this rationale at lines 235-241.

Finally, we only found 2 links between proviral sequences recovered with STIP-Seq at T1/T2 and rebound plasma sequences recovered at T3/T4. Because of this limited number of links and the other reasons mentioned above, we feel shifting focus from T2 to T3 might be complicated in the context of this manuscript.

3. What fraction of the cell clones had integration sites within genes that have been implicated in being able to contribute to cell proliferation? Integration into such genes is well documented but my sense is that it happens only in a small minority of clones yet the authors spend a lot of time on this point. Does the STIP protocol select for these types of integration sites (at least inferred if they are over-represented)?

Out of 22 different clones, 6 (27%) had an integration site in a cancer-related gene/gene involved in cellular proliferation. When compared to integration site datasets on bulk CD4 T cell DNA, the frequency of STIP-Seq integration sites in genes involved in cellular proliferation is not higher: Maldarelli *et al.*, *Science*, 2014 (58%, 17/29) and Wagner *et al.*, *Science*, 2014 (30%, 9/30). This shows that the STIP-Seq protocol probably does not enrich for cells with an integration site in a gene involved in cellular proliferation. This is now mentioned in the revised manuscript at lines 169-173.

4. Please give the reader a little more context for "index sorting" when it first appears in the Results.

We have now added an additional line explaining the basic principle of index sorting in the results section at lines 114-116 as follows: "In addition, the TCR β chain of the host cell was sequenced as described, and the memory phenotype of the cell was determined post hoc by index sorting, during which the level of expression of all phenotypic markers on single-sorted cells was recorded (Fig. 1a, Supplementary Fig. 2)." and in the methods section at lines 492-494 as follows: "Index sorting is a procedure where coordinates of single-sorted cells for all markers are documented, allowing for retrospective determination of the phenotype of each individual sorted cell"

5. The deletions in p17 are curious. Might there be a bias in detecting these proviruses? The loss of p17/MA/membrane targeting may result in an accumulation of p24/CA/Gag in the cytoplasm making this a robust signal.

We thank the reviewer for bringing up this interesting point. To explore this possibility, we examined the fluorescence intensity (FI) of both p24-antibodies (APC and FITC) for each sorted cell. In order to reach sufficient numbers to draw conclusions, FI data from different participants were pooled. To this end, a z-score (standard score) normalization was performed for each experiment before pooling:

$$\text{Normalized FI} = \frac{\text{FI} - \text{mean FI (all p24+ cells of experiment)}}{\text{SD (all p24+ cells of experiment)}}$$

Panel a: correlation dot-plot of normalized FITC and APC FI. A non-parametric Spearman rank correlation test was performed ($\rho = 0.93$, $p < 0.0001$).

Panel b-c: normalized APC FI (panel b) and normalized FITC FI (panel c) of cells harboring a provirus with gag AUG deletion ($n=28$) vs. proviruses with intact gag AUG ($n=99$). A two-sided non-paired Wilcoxon signed-rank test was used to compare both populations ($p=0.71$ for APC and $p=0.70$ for FITC). No significant difference in FI was observed between the 2 groups, suggesting that the lack of an intact p17 gene does not lead to the accumulation of p24 in the cell.

For reference, we performed the same analysis for cells harboring a defective vs. an intact provirus:

Panel d-e: normalized APC FI (panel d) and normalized FITC FI (panel e) of cells harboring a defective provirus vs. an intact provirus. A two-sided non-paired Wilcoxon signed-rank test was used to compare both populations ($p=0.70$ for APC and $p=0.49$ for FITC). No significant difference in FI was observed between the two groups, suggesting that intact proviruses produce similar levels of p24 when compared to defective proviruses upon PMA/i stimulation.

We have added these figures in the revised manuscript as Supplementary Fig. 4, described in the main text at lines 137-143.

Reviewer #2 (Remarks to the Author):

In this study, Cole et al. described a novel single-cell technique, STIP-seq, which allows simultaneous detection of HIV integration sites, proviral genotypes, TCR sequences, and phenotypes of p24-translation-competent HIV-infected cells from clinical samples. Using STIP-seq, the authors examined eight HIV-infected individuals, obtained 158 p24+ cells which yielded 156 integration sites, 40 distinct proviral genomes, and 43 distinct TCR-beta CDR3 sequences. The authors observed that (1) p24+ cells preferentially displayed a memory phenotype and were enriched in proviral genomes with 5' deletions in particular in the MSD stem-loop 2 region, (2) some p24+ cells were associated with TCR sequences specific to M. tuberculosis, influenza and CMV and which had undergone clonal expansion suggested by integration site analyses, and finally (3) plasma viral envelop sequences derived from time points immediately before and after virologic rebound shared sequence identity with STIP-seq-derived sequences and provided evidence that clonally-expanded translation-competent infected cells could have contributed to low level viremia and/or plasma virologic rebound.

Overall, this is a very well-written manuscript and a very well-designed study. Techniques and analyses are sophisticated and robust. Conclusions are valid. STIP-seq is an important advancement in the field of HIV persistence research and enables high-resolution interrogation of the persistent viral reservoir. Perhaps the most important distinction from previous reports and technologies is that STIP-seq enables the authors to link antigen-specificity to viral genome-intactness and translation-competency of the proviral genomes.

We thank the reviewer for the positive evaluation of our manuscript. We addressed the comments in the text below.

Major comments:

1. I am surprised to see that only five out of the 40 distinct proviral genomes detected were genome-intact and many had no Gag ATG and still produced detectable p24. Could this be attributable to PMA/i stimulation leading to abnormal start codon usage?

Preliminary data presented at the Conference on Retroviruses and Opportunistic Infections (CROI, 6-10 March, 2021, abstract #305) by Lee *et al.* suggest that PMA/i stimulation can lead to usage of non-conventional splice sites. Given this observation, it is possible that PMA/i favors the usage of alternative start codons as well. As such, we acknowledge the possibility that proviruses which would not be able to splice or translate under physiological conditions could potentially be picked up by the STIP-Seq assay following PMA/i stimulation. To confirm this hypothesis, supplementary data will need to be generated with latency reversal agents (LRA) other than PMA/i, such as CD3/CD28 or HDAC inhibitors. While we are interested in investigating the differences in proviral species reactivated by different LRAs (as mentioned at lines 420-422), we feel these experiments are out of the scope of the current manuscript. We have now addressed these points to the discussion at lines 334-342.

What was the MFI of the Gag+ populations? Did cells that have Gag ATG displayed higher fluorescence?

This is indeed an interesting hypothesis. We kindly refer to the response to point 5 of Reviewer #1. In this analysis, we show that there is no significant difference in fluorescence intensity between cells harboring a provirus with intact vs. deleted *gag* AUG, suggesting that they produce comparable amounts of p24 protein upon PMA/i stimulation. We have added this figure to the revised manuscript as Supplementary Fig. 4, described in the main text at lines 137-143.

2. Sequencing accuracy is key to genome-intactness versus defectiveness inferences. Please report metrics such as min/max/median sequencing depth per base position and give more details to the de novo assembly step. Were the viral genomes that shared identical integration sites 100% sequence-identical? That will give an estimate of sequencing error.

We agree that sequencing accuracy is key to draw conclusions on genome-intactness, and we acknowledge that our methods description in the original manuscript was not sufficiently detailed. We have now added details on the sequencing, *de novo* assembly, and classification strategy of the NFL genomes. Metrics on sequencing depth and accuracy were added as well. In short, each provirus was sequenced with approximately 200,000 reads per library, yielding a mean sequencing depth of 3767 x with 95% CI (3571, 3964) per base position (after *de novo* assembly). To ensure good coverage, a minimum of 1000 x per-base coverage was set as a threshold. As requested, we also estimated the sequencing error using viral genomes with a shared integration site (clones) as a proxy. Performing mean pairwise distance calculations, we observed a mean per-base error rate of 0.0019%, which is equivalent to 1 error per 52,631 nucleotides sequenced (95% CI: 1 per 132,857 to 1 per 32,461). These additions can be found at lines 538-579.

3. Due to the short V1-V3 env sequence length from the plasma compartment, it is not absolute that the clones identified from STIP-seq are responsible for LLV and/or rebound (as the authors have acknowledged in text). I do agree that it is likely. I would suggest avoiding confirmative language such as “STIP-Seq captures clones that contribute to LLV and viral rebound” (line 274, also line 358, 408, abstract etc).

We agree with the reviewer that links based on V1-V3 *env* might not always be accurate, as described in the ‘limitations’ paragraph of the discussion. As suggested by the reviewer, we have now removed all confirmative language throughout the manuscript.

Minor comments:

1. Line 109. Only 2- and 5- was mentioned. Was the 4-amplicon approach used?

We thank the reviewer for pointing this out. Indeed, the 4-amplicon approach was used on rare occasions, as can be seen in Supplementary Table 2. We have modified the text accordingly (line 112).

2. The original design of the 5-amplicon primer set was actually developed by a Vancouver team Lee GQ AIDS 2017 (PMID: 28832407) for pan-subtype NFL plasma HIV sequencing, and was later validated and applied to MIP-seq Einkauf KB JCI 2019 (PMID: 30688658). Please consider citing in Supp Table 5.

We have added the reference at the appropriate position (line 524).

3. Line 181-182. Did you see identical TCR but non-identical integration sites?

We did not observe identical TCR with different integration sites. We have also uploaded a Source Data file, in which all data per p24+ cell is summarized. From this, the relationship between TCR sequence, phenotype, integration site, p24 fluorescence intensity and proviral genome class can be observed.

4. Line 195. Since antigen specificities were bioinformatically-inferred, these cells were not technically “antigen-responsive”

We changed “antigen-responsive cells” to “cells with predicted TCR-specificity towards a pathogen” (lines 211-213).

5. Line 222. Plasma viral sequencing was not in the methods section. If both T1 and T2 had undetectable viral load, did you use an ultrasensitive approach? Please add to methods section.

We apologize for not including this in the original manuscript. Indeed, ultracentrifugation was used to recover viral RNA from plasma samples. The methodology of the plasma viral sequencing is now added to the methods section (lines 580-598).

6. Line 350. Instead of “could still lead to”, consider “could still be associated with”

We have modified the text accordingly (line 362).

Reviewer: Guinevere Q. Lee

Reviewers' Comments:

Reviewer #2:

Remarks to the Author:

In this revision, Cole et al. has responded to both reviewers' comments diligently and adequately. I have no major comments and support publication of this manuscript. I believe STIP-Seq is an important technological advancement for the HIV persistence research community.

Two minor comments:

Line154-155. Distinct from FLIPs (near full-length viral genome SGA sequencing), the 5-amplicon approach was not designed to capture truncated genomes due to primers binding to the mid portions of the viral genome. This means, by definition MIP-seq biases towards genomes that are at least close to full length (MIP-seq was originally designed to capture genome-intact proviruses). The 4-amplicon and 2-amplicon approaches likely had the same bias but to a lesser extent. Therefore, the inability to detect truncated genomes may not necessarily be related to PMA/i stimulation nor sampling depth. I would suggest editing this sentence.

Rebuttal letter page 6. The authors wrote, "...suggest that PMA/i stimulation can lead to usage of non-conventional splice sites". In fact, the CROI 2021 abstract #305 presentation on HIV transcriptome sequences after PMA/I stimulation suggests that the transcripts resembled truncated HIV DNA genomes (large internal deletions) obtained via FLIPs and the break points did not match known splice junction motifs but there was no direct prove of non-conventional splice site usage. No action is necessary for this comment.

Guinevere Q. Lee

REVIEWERS' COMMENTS

Reviewer #2 (Remarks to the Author):

In this revision, Cole et al. has responded to both reviewers' comments diligently and adequately. I have no major comments and support publication of this manuscript. I believe STIP-Seq is an important technological advancement for the HIV persistence research community.

Two minor comments:

Line154-155. Distinct from FLIPs (near full-length viral genome SGA sequencing), the 5-amplicon approach was not designed to capture truncated genomes due to primers binding to the mid portions of the viral genome. This means, by definition MIP-seq biases towards genomes that are at least close to full length (MIP-seq was originally designed to capture genome-intact proviruses). The 4-amplicon and 2-amplicon approaches likely had the same bias but to a lesser extent. Therefore, the inability to detect truncated genomes may not necessarily be related to PMA/i stimulation nor sampling depth. I would suggest editing this sentence.

We acknowledge that the 5-amplicon approach was designed to capture close to full-length proviruses, and would eventually miss truncated proviruses. However, the sentence at lines 154-155 refers to the inability of STIP-Seq to pick up two **intact** (non-truncated) proviruses that were detected by FLIPs at this timepoint, rather than truncated proviruses. Indeed, in the sentence preceding this, we mention that with FLIPs (1-amplicon approach), two intact proviruses were identified on bulk CD4 T cell DNA. However, these same proviruses were not recovered with STIP-Seq from CD4 T cells at the same timepoint. This showed that either these proviruses were not reactivated with PMA/ionomycin, or they were missed because the sampling depth was lower with STIP-Seq. As such, we think the sentence remains correct. However, we do think that the reviewer raises an interesting point about the potential bias of multi-amplicon approaches. We have acknowledged this at lines 153-156, as follows: "We acknowledge the possibility that the number of truncated proviruses recovered by STIP-Seq might be underestimated since we used multi-amplicon approaches to amplify the proviral genomes, which could preclude the detection of truncated proviruses that contain deletions in the primer binding sites regions".

Rebuttal letter page 6. The authors wrote, "...suggest that PMA/i stimulation can lead to usage of non-conventional splice sites". In fact, the CROI 2021 abstract #305 presentation on HIV transcriptome sequences after PMA/I stimulation suggests that the transcripts resembled truncated HIV DNA genomes (large internal deletions) obtained via FLIPs and the break points did not match known splice junction motifs but there was no direct prove of non-conventional splice site usage. No action is necessary for this comment.

Guinevere Q. Lee